# Vision Search Assistant: Empower Vision-Language Models as Multimodal Search Engines

## Abstract

Search engines enable the retrieval of unknown information with texts. However, traditional methods fall short when it comes to understanding unfamiliar visual content, such as identifying an object that the model has never seen before. This challenge is particularly pronounced for large vision-language models (VLMs): if the model has not been exposed to the object depicted in an image, it struggles to generate reliable answers to the user's question regarding that image. Moreover, as new objects and events continuously emerge, frequently updating VLMs is impractical due to heavy computational burdens. To address this limitation, we propose Vision Search Assistant, a novel framework that facilitates collaboration between VLMs and web agents. This approach leverages VLMs' visual understanding capabilities and web agents' real-time information access to perform open-world Retrieval-Augmented Generation via the web. By integrating visual and textual representations through this collaboration, the model can provide informed responses even when the image is novel to the system. Extensive experiments conducted on both open-set and closed-set QA benchmarks demonstrate that the Vision Search Assistant significantly outperforms the other models and can be widely applied to existing VLMs.

## 1 Introduction

The advent of Large Language Models (LLMs) (Achiam et al., 2023; OpenAI, 2024; Anthropic, 2024; Touvron et al., 2023; Schulman et al., 2022; Chiang et al., 2023) has significantly enhanced the human capacity to acquire unfamiliar knowledge through powerful zero-shot Question-Answering (QA) capabilities. Building upon these advancements, techniques such as Retrieval-Augmented Generation (RAG) (Yu et al., 2023; Shi et al., 2023; Trivedi et al., 2022) have further reinforced LLMs in knowledge-intensive, open-domain QA tasks. Concurrently, recent progress in visual instruction tuning (Liu et al., 2023b;a; Zhu et al., 2023) has led to the development of large Vision-Language Models (VLMs) that aim to equip LLMs with visual understanding capabilities. By scaling model parameters and training on extensive text-image datasets, VLMs such as LLaVA-1.6-34B (Liu et al., 2023a), Qwen2-VL-72B (Bai et al., 2023), and InternVL2-76B (Chen et al., 2024b) have achieved state-of-the-art performance on the OpenVLM leaderboard[1]. However, LLMs and VLMs are subject to the limitations imposed by their knowledge cutoff dates. They may provide incorrect answers when asked about events or concepts that occurred after their knowledge cutoff dates (Figure 1) To overcome this limitation for LLMs, they are often connected to web agents (Liu et al., 2023d; Nakano et al., 2021; Chen et al., 2024a; Deng et al., 2024; Bai et al., 2024), which enable internet access and information retrieval, allowing them to obtain the most up-to-date data and improve the accuracy of their responses. Such agents are designed to interpret natural language instructions, navigate complex web environments, and extract relevant textual information from HTML documents, thereby enhancing the accessibility and utility of vast amounts of web-based textual data for a wide range of applications.

*However, for VLMs facing unseen images and novel concepts, their ability to learn and use up-to-date multimodal knowledge from the internet remains a pressing challenge.* As the existing web

---

[1] https://huggingface.co/spaces/opencompass/open_vlm_leaderboard

Figure 1: **Vision Search Assistant acquires unknown visual knowledge through web search**. An intuitive comparison of answering the user's question with an unseen image. The proposed Vision Search Assistant is developed based on LLaVA-1.6-7B, and its ability to answer the question on unseen images outperforms the state-of-the-art models including LLava-1.6-34B (Liu et al., 2023a), Qwen2-VL-72B (Bai et al., 2023), and InternVL2-76B (Chen et al., 2024b).

agents predominantly rely on searching the user's question and summarizing the returned HTML content, they present a notable limitation when handling tasks involving images or other visual content: the visual information is often overlooked or inadequately processed.

In this work, we enable the VLM to answer a question regarding an unseen image or novel concept, which behaves like a human searching the Internet. It **1)** understands the query, **2)** decides which objects in the image it should look at and infers the correlations among the objects, **3)** respectively generates the texts to search, **4)** analyzes the contents returned from the search engine based on the query and inferred correlations, and **5)** judges if the obtained visual and textual information is sufficient for generating the answer or it should iterate and refine the above process. Regarding the concrete designs of such a framework, we make contributions by answering the following three questions that remained unanswered in the literature

- *What to search*: shall we search the descriptions of the whole image or some critical objects?
- *How to search*: shall we search once and summarize a huge amount of returned content or search progressively to obtain more related content?
- *By what to conclude*: shall the final answer be generated with the eventually summarized web knowledge or all the knowledge acquired through the entire searching process?

By exploring such three aspects, we propose **Vision Search Assistant**, a framework based on VLM-agent collaboration, which *empowers an arbitrary VLM to become a multimodal automatic search engine*. We integrate VLMs into web agents to understand what the user wants, where to look, what to search, how to learn from the returned multimodal information, and whether to conclude or search another time. More specifically, Vision Search Assistant conducts three steps (Figure 3):

- Visual Content Formulation (§ 3.1) is proposed to represent the visual content with VLM-generated textural descriptions of critical visual objects and their underlying correlations. Through this step, we obtain a *correlated formulation* for each critical object, which is a textual representation that considers its correlations with other objects.

Figure 2: **Comparsion with Closed-Source Models including GPT-4o (OpenAI, 2024), Gemini (Reid et al., 2024), Claude 3.5 Sonnet (Anthropic, 2024) with Vision Search Assistant** shows that Vision Search Assistant satisfies users' needs better even if the image is novel.

- Web Knowledge Search (§ 3.2) is a novel algorithm that drives the search process. It generates multiple sub-questions with the web agent regarding the user prompt and the correlated formulation of each critical object. Each of such sub-questions can be viewed as a node in a directed graph. For each correlated formulation and each sub-question, we construct the search query by combining the correlated formulation and sub-question and use the LLM to analyze and select useful contents returned from the search engine, then summarize the web knowledge from the answers obtained with all such sub-questions. After that, we iterate the above step by proposing more sub-questions based on the previous sub-questions and known web knowledge, which can be seen as expanding the directed graph. We use the LLM to judge if the latest iteration has obtained sufficient web knowledge to answer the user's question and terminate the process if so.
- Collaborative Generation (§ 3.3)) is proposed to use the VLM to generate the eventual answer with all the critical objects in the image, the initial question, all of their correlated formulations, and the web knowledge obtained in every iteration.

As shown in Figure 2, Vision Search Assistant can generate more precise answers than powerful closed-source models such as GPT-4o (OpenAI, 2024), Gemini (Reid et al., 2024), and Claude 3.5 Sonnet (Anthropic, 2024), which further validates the necessity and promising improvement of VLM-Agents collaboration in tackling the growing complexity of multimodal web data and the rapid influx of novel visual content.

## 2 RELATED WORK

**Vision-Language Models**. Pioneering models such as Flamingo (Alayrac et al., 2022), BLIP-2 (Li et al., 2023), LLaVA (Liu et al., 2023b), and MiniGPT-4 (Zhu et al., 2023) have been instrumental in training vision-language models for the tasks such as image captioning and visual question answering. Recent works focus on higher-quality datasets (Gong et al., 2023) and developing lightweight, trainable models (Gao et al., 2023) to enhance efficiency and accessibility. Further progress includes extending large language models (LLMs) to additional modalities and domains, such as audio processing (Huang et al., 2023; Chen et al., 2023a), and more modalities (Han et al., 2024). Additionally, KOSMOS-2 (Peng et al., 2023), InternVL2 (Chen et al., 2024b), MiniGPT-2 (Chen et al., 2023b), and LLaVA-1.5 (Liu et al., 2023a) incorporate region-level information by encoding visual regions to embeddings of language models. However, despite scaling model parameters and training data, VLMs' ability to handle unseen images remains limited, as they heavily rely on previously seen text-image pairs. To overcome this, we propose to enhance VLMs' performance on novel data by improving generalization without relying solely on extensive training pairs.

**Web Search Agents**. The development of web search agents has progressed through integrating advanced learning techniques, enhancing autonomy, and optimizing efficiency in web automation. Early models like WebGPT (Nakano et al., 2021) and WebGLM (Liu et al., 2023d) primarily focused on retrieving information for question-answering tasks, while newer models, such as AutoWebGLM (Lai et al., 2024), address deployment challenges with compact designs. Despite their strong web navigation skills, larger models such as WebAgent (Gur et al., 2023) are constrained by size. Incorporating reinforcement learning (Bai et al., 2024) and behavior cloning (Zheng et al., 2024; Patel et al., 2024) has further boosted the efficiency of web agents, as demonstrated by MindAct (Deng et al., 2024), which integrates cognitive functionalities for complex task execution. While these advances are leading to more scalable and versatile solutions for real-world use, current web agents still struggle with processing visual content directly from the web. We introduce Vision-Language Models to enable web agents to effectively interpret and interact with visual data, significantly expanding their capabilities in handling complex, multimodal tasks. We hope it can make web agents more powerful and adaptable in real-world applications.

**Retrieval-Augmented Generation**. Integrating retrieval from large corpora into language models has become essential for knowledge-intensive tasks like open-domain question answering. Instead of relying solely on pre-trained data, the retriever-reader architecture (Chen et al., 2017; Guu et al., 2020) enables models to fetch relevant information based on an input query, which the language model then uses to generate accurate predictions. Recent research has enhanced retrievers (Karpukhin et al., 2020; Xiong et al., 2020a; Qu et al., 2020; Xiong et al., 2020b; Khalifa et al., 2023), improved readers (Izacard & Grave, 2020b; Cheng et al., 2021; Yu et al., 2021; Borgeaud et al., 2022), jointly fine-tuned both components (Yu, 2022; Izacard et al., 2022; Singh et al., 2021; Izacard & Grave, 2020a), and integrated retrieval directly within language models (Yu et al., 2023; Shi et al., 2023; Trivedi et al., 2022).

Therefore, we propose the Vision Search Assistant framework, which introduces an open-world retrieval-augmented generation framework that extends beyond text-based retrieval to operate across both vision and language modalities on the web. It enables VLMs to access real-time, dynamic information, improving their ability to handle novel, cross-modal queries. By pushing the boundaries of retrieval beyond static knowledge sources, we address the challenge of incorporating web-based, multimodal data into generative tasks, offering a more adaptable and scalable solution for RAG.

## 3 VISION SEARCH ASSISTANT

### 3.1 VISUAL CONTENT FORMULATION

The Visual Content Formulation is proposed to extract the object-level descriptions and correlations among objects in an image. Given the input image $X_I$, we first use the open-vocab detector $\mathcal{F}_{\det}(\cdot)$ (Liu et al., 2023c) to obtain $N$ regions of interests in the original image,

$$\{X_I^{(i)}\}_{i=1}^N = \mathcal{F}_{\det}(X_I), \tag{1}$$

where $i$ indicates the $i$-th region $X_I^{(i)}$ in the image $X_I$. Then we employ the pretrained VLM [2] $\mathcal{F}_{\text{vlm}}(\cdot, \cdot)$ to caption these regions $\{X_I^{(i)}\}_{i=1}^N$ conditioned on the tokenized user's textual prompt $X_T$, and obtain the visual caption $X_r^{(i)}$ for the $i$-th region:

$$X_r^{(i)} = \mathcal{F}_{\text{vlm}}(X_I^{(i)}, X_T). \tag{2}$$

In this way, we make the regional captions $\{X_r^{(i)}\}_{i=1}^N$ contain specific visual information obtained based on the user's interests. To formulate the visual content more comprehensively, we further correlate these visual regions to obtain precise descriptions of the whole image. More specifically, for each region, we concatenate its corresponding caption and the captions of all the other regions. The resultant text, denoted by $[X_r^{(i)}, \{X_r^{(j)}\}_{j \neq i}]$, encodes the underlying correlations. It is fed into

---

[2] Our experiments are conducted with LLaVA-1.6-Vicuna-7B model, which is publicly available at `https://huggingface.co/liuhaotian/llava-v1.6-vicuna-7b`.

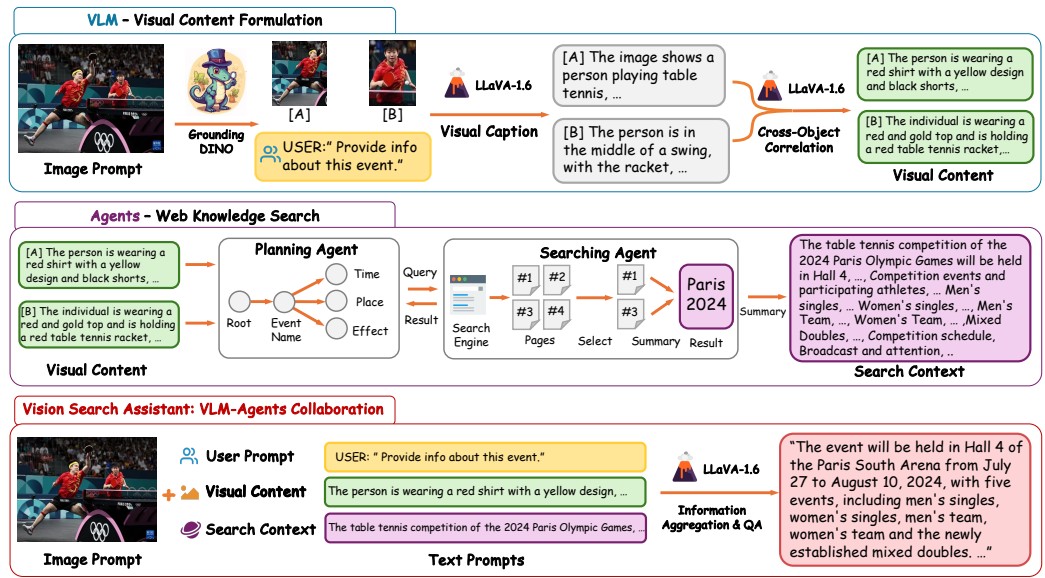

Figure 3: **Overview of Vision Search Assistant**. We first identify the critical objects and generate their descriptions considering their correlations, named Correlated Formulation, using the Vision Language Model (VLM). We then use the LLM to generate sub-questions that leads to the final answer, which is referred to as the Planning Agent. The web pages returned from the search engine are analyzed, selected, and summarized by the same LLM, which is referred to as the Searching Agent. We use the original image, the user's prompt, the Correlated Formulation together with the obtained web knowledge to generate the final answer. Vision Search Assistant produces reliable answers, even for novel images, by leveraging the collaboration between VLM and web agents to gather visual information from the web effectively.

the VLM together with the image region $\boldsymbol{X}_I^{(i)}$. The output is referred to as the *correlated formulation* of each region $\{\boldsymbol{X}_c^{(i)}\}_{i=1}^N$.

$$\boldsymbol{X}_c^{(i)} = \mathcal{F}_{\text{vlm}}(\boldsymbol{X}_I^{(i)}, [\boldsymbol{X}_r^{(i)}, \{\boldsymbol{X}_r^{(j)}\}_{j\neq i}]). \tag{3}$$

We will use the correlated formulations of such regions to perform the following web search.

## 3.2 WEB KNOWLEDGE SEARCH: THE CHAIN OF SEARCH

The core of Web Knowledge Search is an iterative algorithm named *Chain of Search*, which is designed to obtain the comprehensive web knowledge of the correlated formulations $\{\boldsymbol{X}_c^{(i)}\}_{i=1}^N$. We take an arbitrary $i$-th region $\boldsymbol{X}_c^{(i)}$ to elaborate on the Chain of Search algorithm and drop the superscript $(i)$ for convenience.

We use the LLM in our VLM to generate sub-questions that lead to the final answer, which is referred to as the Planning Agent. The web pages returned from the search engine are analyzed, selected, and summarized by the same LLM, which is referred to as the Searching Agent. In this way, we can obtain web knowledge regarding the visual content. Then, based on each of such sub-questions, the Planning Agent generates more sub-questions, and the Searching Agent obtains web knowledge for the next iteration. Formally, we define a directed graph to represent this process, which is $\mathcal{G} = \langle V, E \rangle$, where $V = \{V_0\}$ is the set of nodes, $V_0$ is the initial node, and $E = \varnothing$ is the set of edges. A node represents a set of known information so that $V_0$ should represent what we know about the region before any web search, i.e., the correlated formulation $\boldsymbol{X}_c$. This is formulated as $V_0 \leftarrow \boldsymbol{X}_c$. When we search with a sub-question, we will update the graph with a new node representing the web knowledge gained through the sub-question.

For the 1-st update, we generate sub-questions based on $V_0$ and denote the generated sub-questions by $(\boldsymbol{X}_s^1) = \{(\boldsymbol{X}_s^1)_i\}_{i=1}^{N_v^1}$, where $N_v^1$ is the number of sub-questions, i.e., the number of new nodes.

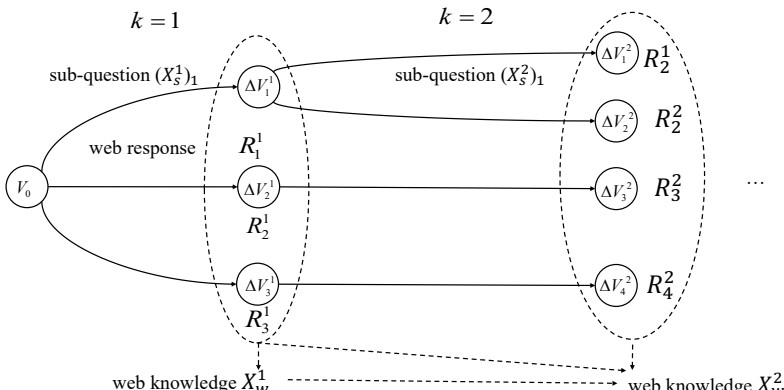

Figure 4: **The Chain of Search algorithm (§ 3.2)**. We deduce the update of the directed graph when $k = 1, 2, \cdots$, and the web knowledge is progressively extracted from each update.

Let $j$ be the index of the sub-question, the new node $\Delta V_j^{(1)}$ is a child of $V_0$, which corresponds to a search with sub-question $(\boldsymbol{X}_s^1)_j$. The returned set of web pages are formatted as HTML documents. The Searching Agent uses the LLM in our VLM, which is denoted by $\mathcal{F}_{llm}(\cdot)$, to judge their relevance to the parent node $V_0$ and the corresponding sub-question $(\boldsymbol{X}_s^1)_j$ and select those of the highest relevance. The selected web page index $\tau_j^1$ can be formulated

$$\tau_j^1 = \mathcal{F}_{\text{llm}}([V_0, (\boldsymbol{X}_s^1)_j]). \tag{4}$$

We use $\tau_j^1$ to select a subset of the HTML documents at the 1-st update, and those selected for sub-question $j$ are denoted by $\{P_j^1\}$. We derive the *search response* $R_j^1$ for sub-question $j$ at the 1-st update by summarizing the selected pages with the LLM, which is $R_j^1 = \mathcal{F}_{\text{llm}}(\{P_j^1\})$. By the definition of the directed graph, the new node $\Delta V_j^{(1)}$ should represent $R_j^1$, that is, $\Delta V_j^{(1)} \leftarrow R_j^1$. We add $\Delta V_j^{(1)}$ into the node set and $(V_0, V_j^{(1)})$ into the edge set. In this paper, $\Delta V_j^{(1)}$ is synonymous with "the search response $R_j^1$ obtained with sub-question $(\boldsymbol{X}_s^1)_j$".

Then, we summarize the search responses of all the $N_v^1$ nodes at the 1-st update and obtain the comprehensive *web knowledge* $\boldsymbol{X}_w^{(1)}$, which is denoted by

$$\boldsymbol{X}_w^{(1)} = \mathcal{F}_{\text{llm}}([R_1^1, R_2^1, \cdots, R_{N_v^1}^1]). \tag{5}$$

For the following updates with $k > 1$, we expand the graph similarly but with minor differences:

- For each node at update $k - 1$, we use the LLM to generate further sub-questions, just like how we expand $V_0$ at the 1-st update.

- When we select the most relevant web pages for a node $\Delta V_j^{(k)}$, we analyze their relevance to not only $V_0$ and the corresponding sub-question $(\boldsymbol{X}_s^k)_j$ (just like the 1-st update), but also the search response of its parent node.

- When we summarize the comprehensive web knowledge $\boldsymbol{X}_w^{(k)}$, except for the search responses of all the nodes at the current update, we also use all the known comprehensive web knowledge $\{\boldsymbol{X}_w^{(i)}\}_{i=1}^{k-1}$ and the search responses of all the previous nodes $\{R_m^n\}_{\{m=1,n=1\}}^{\{m=N_v^n, n=k-1\}}$.

Formally,

$$\begin{aligned} \tau_j^k &= \mathcal{F}_{\text{llm}}([V_0, (\boldsymbol{X}_s^k)_j, R_i^{k-1}]), \\ \boldsymbol{X}_w^{(k)} &= \mathcal{F}_{\text{llm}}(\{\boldsymbol{X}_w^{(i)}\}_{i=1}^{k-1}, \{R_m^n\}_{\{m=1,n=1\}}^{\{m=N_v^n, n=k-1\}}, \{R_i^k\}_{i=1}^{N_v^k}), \ k > 1. \end{aligned} \tag{6}$$

At each update, the search agent uses the LLM to judge if the knowledge currently obtained is sufficient to answer the initial question. If so, we terminate the process.

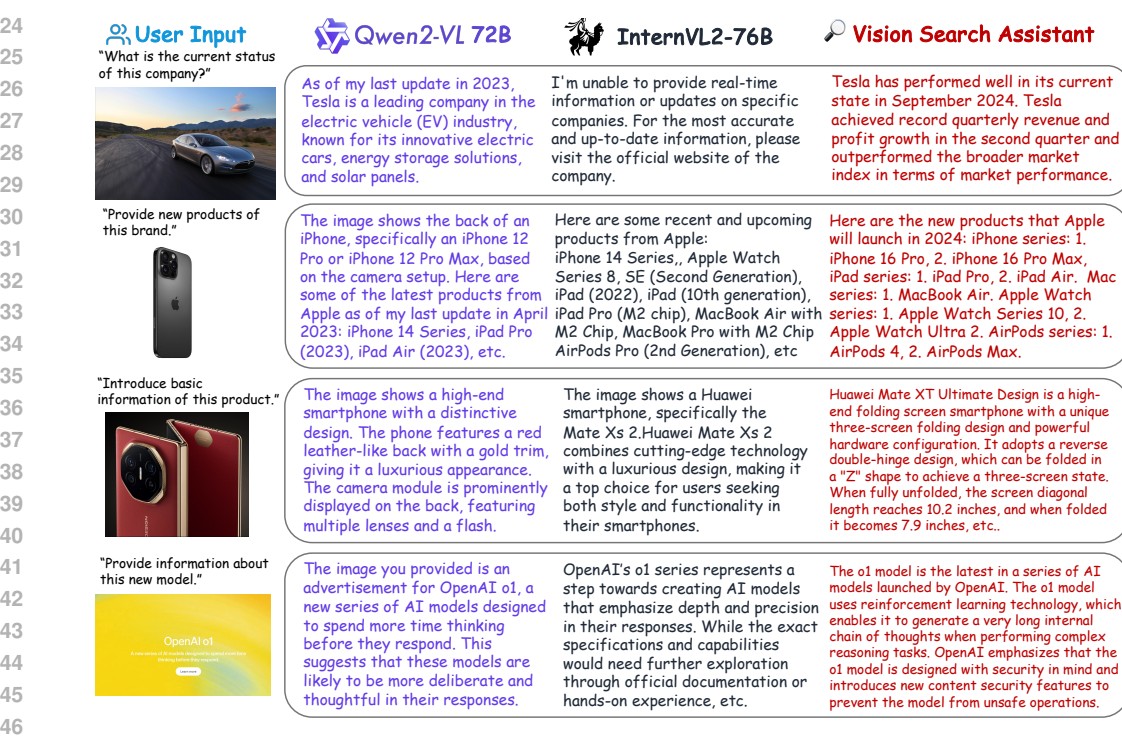

Figure 5: **Comparisons among Qwen2-VL-72B, InternVL2-76B, and Vision Search Assistant.** We compare the open-set QA results on both novel events (the first two rows) and images (the last two rows). Vision Search Assistant excels in generating accurate and detailed results.

## 3.3 COLLABORATIVE GENERATION

We use the original image $\boldsymbol{X}_I$, the user's initial prompt $\boldsymbol{X}_T$, and the Correlated Formulations $\{\boldsymbol{X}_C^{(i)}\}_{i=1}^N$ together with the obtained web knowledge $\{\boldsymbol{X}_W^{(i)}\}_{i=1}^N$ to collaboratively generate the final answer $\boldsymbol{Y}$ with the VLM:

$$\boldsymbol{Y} = \mathcal{F}_{\mathrm{vlm}}(\boldsymbol{X}_I, \{\boldsymbol{X}_c^{(i)}\}_{i=1}^N, \{\boldsymbol{X}_w^{(i)}\}_{i=1}^N, \boldsymbol{X}_T). \tag{7}$$

## 4 EXPERIMENTS

### 4.1 OPEN-SET EVALUATION

**Setup**. In the Open-Set Evaluation, we performed a comparative assessment by 10 human experts evaluation, which involved questions of 100 image-text pairs collected from the news from July 15th to September 25th covering all fields on both novel images and events. Human experts conducted the evaluations across three critical dimensions: factuality, relevance, and supportiveness.

**Results and Analysis**. As illustrated in Figure 6, Vision Search Assistant demonstrated superior performance across all three dimensions compared to Perplexity.ai Pro and GPT-4-Web: **1**) Factuality: Vision Search Assistant scored 68%, outperforming Perplexity.ai Pro (14%) and GPT-4-Web (18%). This significant lead indicates that Vision Search Assistant consistently provided more accurate and fact-based answers. **2**) Relevance: With a relevance score of 80%, Vision Search Assistant demonstrated a substantial advantage in providing highly pertinent answers. In comparison, Perplexity.ai Pro and GPT-4-Web achieved 11% and 9%, respectively, showing a significant gap in their ability to maintain topicality with the web search. **3**) Supportiveness: Vision Search Assistant also outperformed the other models in providing sufficient evidence and justifications for its responses, scoring 63% in supportiveness. Perplexity.ai Pro and GPT-4-Web trailed with scores of 19% and 24%, respectively. These results underscore the superior performance of Vision Search Assistant

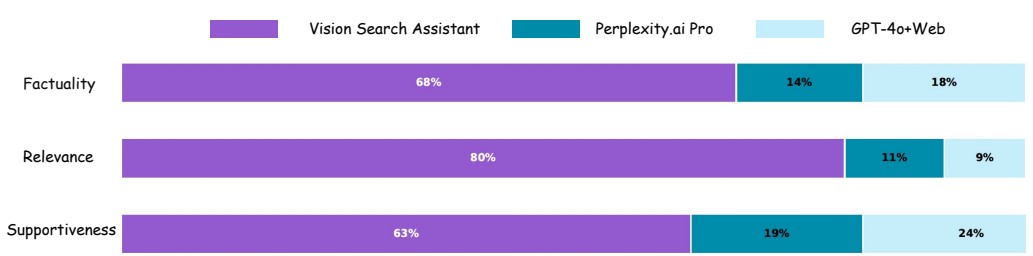

Figure 6: **Open-Set Evaluation**: We conduct a human expert evaluation on open-set QA tasks. Vision Search Assistant significantly outperformed Perplexity.ai Pro and GPT-4o-Web across three key objectives: factuality, relevance, and supportiveness.

in open-set tasks, particularly in delivering comprehensive, relevant, and well-supported answers, positioning it as an effective method for handling novel images and events.

## 4.2 CLOSED-SET EVALUATION

**Setup**. We conduct the closed-set evaluation on the LLaVA-W Liu et al. (2023a) benchmark, which contains 60 questions regarding the Conversation, Detail, and Reasoning abilities of VLMs in the wild. We use the GPT-4o(0806) model for evaluation. We use LLaVA-1.6-7B as our baseline model, that has been evaluated in two modes: the standard mode and a "naive search" mode that utilizes a simple Google Image search component. Additionally, an enhanced version of LLaVA-1.6-7B, equipped with improvements outlined in section § 3.2, is also evaluated.

| Model | Conversation (%) | Detail (%) | Reasoning (%) | Overall (%) |
|---|---|---|---|---|
| LLava-1.6-7B (Baseline) | 72.9 | 76.5 | 84.2 | 78.5 |
| LLava-1.6-7B (naive search) | 70.3 | 76.7 | 85.8 | 78.9 |
| LLava-1.6-7B (*w/* § 3.2) | 72.6 | 78.9 | 89.8 | 82.7 |
| Vision Search Assistant | **73.3** (+0.4) | **79.3** (+2.8) | **95.0** (+10.8) | **84.9** (+6.4) |

Table 1: **Closed-Set Evaluation on the LLaVA-W benchmark**. We use GPT-4o (0806) for evaluation. Naive search here denotes the VLM with Google image search.

**Results and Analysis**. As shown in Table 1, the Vision Search Assistant demonstrates the strongest performance across all categories. Specifically, it achieves a 73.3% score in the conversation category, representing a modest gain of +0.4% compared to the LLaVA models. In the detail category, the Vision Search Assistant stands out with a score of 79.3%, outperforming the highest-performing LLaVA variation by +2.8%. When it comes to reasoning, our method brings out +10.8% above the best-performing LLaVA model. This suggests that the Vision Search Assistant's advanced integration of visual and textual search greatly enhances its reasoning capabilities. The overall performance of the Vision Search Assistant is 84.9%, marking an improvement of +6.4% over the baseline model. This shows that the Vision Search Assistant excels in both conversational and reasoning tasks, giving it a clear advantage for in-the-wild abilities.

## 4.3 ABLATION STUDY

**What to search: Object-Level Descriptions**. As illustrated in Figure 7, if we use the image-based caption, the search agent can not precisely focus on the key information (the handbag in this figure), meanwhile, the image contains visual redundancy, which obstacles the textual description to drive web agent and retrieve the most relevant web pages, therefore, we use the object-level description in the following ablation study.

**Complex Scenarios of Search: Visual Correlation**. We find that the caption can not fully support the search ability in multiple-object scenarios. As shown in Figure 8, the caption of Biden can not answer the questions on the group-wise debate, the visual correlation ("debate" in this demo) between Trump can effectively improve the answer quality.

"Provide information about this type of product this year."

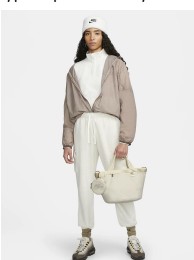

**[Image-Based]:** The image shows a person wearing a beige tracksuit with a hoodie and matching shoes, along with a beige tote bag. The tracksuit is from the adidas brand, specifically the Adicolor Firebird Loose Track Pants series. The hoodie is oversized and has a high neckline. The shoes are low-top with a lace-up front and yellow accents. The style of the image is a fashion or lifestyle photograph, commonly used for showcasing clothing and accessories. The tracksuit is made of recycled polyester and has a wide fit. The hoodie has a classic design with a front pocket and a drawstring closure. The shoes are likely from brands such as Vans, ASOS DESIGN, or ASOS, and are made of soft, possibly fleece-like material. The tote bag has a visible logo on the front and is likely from brands such as Michael Kors or Fossil.

**[Object-Based]:** According to the information you provided, the main style features of Nike handbags in 2024 include: 1. Brand iconic elements: Nike handbags will continue to emphasize brand iconic elements, such as Swoosh patterns. 2. Multifunctional design: Nike handbags focus on practicality and provide multiple functions and designs. 3. Innovative materials and designs: Nike will launch handbags with innovative materials and designs, such as Le Swoosh, a handbag inspired by the Nike Swoosh logo. 4. Combination of fashion and function: Nike handbags will continue to emphasize the combination of fashion and function. 5. Market performance: Nike's handbag series in 2024 has received positive feedback from the market. In terms of popular colors, the popular colors in 2024 are mainly concentrated in the five major colors of gray, cream yellow, red, green and baby blue. In terms of price, the price range of Nike handbags ranges from a few hundred Hong Kong dollars to a few thousand Hong Kong dollars. The specific price is affected by many factors, including region, sales channel and the specific model and design of the handbag.

Figure 7: **Ablation Study on What to Search**. We use the object description to avoid the visual redundancy of the image.

What is Biden debating for in this picture?

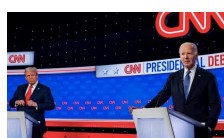

**[Naive Caption]:** Based on the description provided, the man in the image is described as wearing a formal suit and tie and standing at a podium, which may indicate that he is a politician. However, due to the lack of specific visual information, his exact identity cannot be determined. Regarding what he is debating, it is impossible to determine the specific content of his debate because no specific debate topic or context information is provided. Usually, politicians discuss topics such as policy, economy, and social issues during debates. In summary, due to the lack of specific visual information and context information, it is impossible to determine who the man in the image is or what he is debating. If more accurate information is needed, a more detailed description or image link may be required.

**[Visual Correlation]:** In this image, Joe Biden participates in a presidential debate. Biden's positions and topics in the debate mainly focused on the following aspects: 1. COVID-19 Epidemic: Biden emphasized the seriousness of the new crown epidemic, criticized Trump for downplaying the seriousness of the epidemic, and pointed out that Trump's actions failed to effectively control the epidemic, resulting in a large number of American deaths. 2. Medical Insurance: Biden wants to expand coverage and make Obamacare more affordable. He supports a nationwide mandate to wear masks and wants to raise the minimum wage to $15 an hour. 3. Economy: Biden supports increasing the federal minimum wage and hopes to stimulate the economy through increased public services and investment. 4. National Security: Biden promised to take action against any country that interferes in the US election and accused Trump of failing to take action against Russia's actions in terms of election interference. 5. Climate Change: Biden supports joining the Paris Agreement on Climate Change and is committed to protecting the environment.

Figure 8: **Ablation Study on Complex Scenarios**. We use the visual correlation to improve the ability in multiple-object scenarios.

**How to search**: **Chain of Search** (§ 3.2). The trivial idea to incorporate web search with VLMs is to introduce a Google search engine and re-rank the large-scale related pages. As shown in Figure 9, we found it difficult to directly obtain the required knowledge since the page-rank method prefers more hyper-link pages instead of exact relevance. The VLM is also limited to its context length to summarize a large number of pages. Therefore, we propose the chain of search and enable the progressive summary of web knowledge aiming to answer the user's questions.

"List some best papers in the conference where this paper was published.."

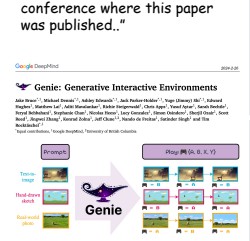

**[Naive Re-rank Web pages with Google]:** Here are some high-quality papers published in the 2024 International Conference on Machine Learning (ICML): 1. "Deep Learning for Graph Neural Networks: A Survey" - Author: Jian Zhang, Ying Sun, Xiao Wang, Jianfeng Gao. 2. "Learning to Optimize: A Survey and Outlook" - Author: Jian Zhang, Ying Sun, Xiao Wang, Jianfeng Gao. Abstract: This paper reviews the research progress of learning optimization algorithms, including gradient descent, stochastic gradient descent, Adam, etc. 3. "Generative Adversarial Networks: A Survey" - Author: Jian Zhang, Ying Sun, Xiao Wang, Jianfeng Gao. Abstract: This paper reviews the research progress of generative adversarial networks (GANs), including the basic principles, variants, applications, and future research directions of GANs. It provides a comprehensive overview of the field of GANs.

**[Chain of Search]:** Based on the information you provided, here are some papers that received high recognition at the ICML 2024 conference: 1. Probabilistic Inference in Language Models via Twisted Sequential Monte Carlo: This paper explores how to use sequential Monte Carlo methods to perform probabilistic inference on large language models and proposes a contrasting method for learning twisted functions. 2. Stealing part of a production language model : This paper first proposed a model stealing attack against black-box production language models, which can accurately extract information from models like OpenAI's ChatGPT or Google's PaLM-2. 3. Scaling Rectified Flow Transformers for High-Resolution Image Synthesis: This paper studies how to improve noise sampling techniques to improve the efficiency of training rectangular flow variant models and proposes a new transformer-based architecture for text-to-image generation. These papers show the latest research results and innovative directions of the ICML 2024 conference in the field of machine learning and artificial intelligence.

Figure 9: **Ablation Study on How to search**. We propose the *Chain of Search* (§ 3.2) to progressively obtain related web knowledge for VLMs.

## 5 CONCLUSION AND DISCUSSION

In this paper, we seek to improve the generalization ability of VLMs of novel images and extend the capacity of web agents to solve visual content tasks. Through the synergistic collaboration between VLMs and web agents, we find that VLMs can generate more reliable answers regarding novel images with the help of real-time web knowledge retrieval, and web agents can solve more challenging tasks than HTML documents only. Meanwhile, there are also some limitations inside the Vision Search Assistant framework such as the exact inference speed of VLMs, the web condition of web agents, and the retrieval efficiency. We hope this paper can inspire more research to address the challenges of VLMs in user experience and improve the automation abilities of web agents across diverse modalities.

"What award did this paper win?"

The paper titled "Generative Image Dynamics" by Zhengqi Li, Richard Tucker, Noah Snavely, and Aleksander Holynski won the Best Paper Award at the IEEE/CVF Conference on Computer Vision and Pattern Recognition (CVPR) in 2024. This award is a significant recognition of the paper's innovative approach to modeling natural oscillation dynamics from a single still picture and its ability to produce photo-realistic animations

"What award did this paper win in 2024?"

In 2024, the paper "Auto-Encoding Variational Bayes" co-authored by Diederik P. Kingma and Max Welling won the Test of Time Award at the International Conference on Computation and Learning Theory (ICLR). "Auto-Encoding Variational Bayes" won this award for its pioneering contributions to the field of variational autoencoders (VAEs), especially its innovative methods for efficiently handling complex data distributions and optimizing model parameters.

"Provide new information about this company in 2024."

1. Google I/O 2024: Google I/O 2024 will be held in San Francisco, USA on May 15, 2024. This conference showcased Google's latest technological progress and innovation in development tools in the fields of artificial intelligence, Web, mobile, and cloud computing. 2. Financial: In the second quarter of 2024, Alphabet (Google's parent company)'s revenue was US$84.742 billion, a year-on-year increase of 14%. Google will continue to play its role as the world's leading search engine and technology company in 2024.

"What's price of this product of Chang 'An?"

The price of the Changan CS75 Plus SUV varies depending on the model and configuration:1. 2024 Changan CS75 Plus Third Generation Champion Edition 1.5T Zhijia Jinghang Type: The suggested retail price ranges from 11.79 to 14.99 million yuan. 2. 2024 Changan CS75 Plus Third Generation Champion Edition 2.0T Zhijia Jingqing Type: The suggested retail price is 13.99 million yuan. 3. 2023 Changan CS75 Plus Third Generation 1.5T Automatic Luxury Type: The suggested retail price is 12.49 million yuan.

"Provide information about new products of this brand of potato chips in 2024."

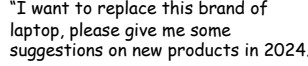
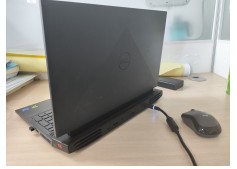

Here is a brief introduction to the new products that Lays will launch in 2024: 1. Wavy Tzatziki: This potato chip is wavy in shape and uses traditional Greek ingredients such as yogurt, lemon, dill and garlic to bring a unique Mediterranean flavor. 2. Masala: This potato chip combines traditional Indian spices such as cumin, coriander, turmeric and pepper to bring a spicy and rich taste. 3. Honey Butter: This potato chip combines the Korean honey and butter flavors to bring a unique sweet but not greasy taste.

"I want to replace this brand of laptop, please give me some suggestions on new products in 2024."

Here are some recommendations for Dell laptops in 2024:1. Dell XPS 14 2024: This laptop has a 16-core Intel Core Ultra 7 or Intel Core Ultra 9 processor, optional integrated Intel Arc graphics or NVIDIA GeForce RTX 40 series graphics, supports Wi-Fi 6E and Bluetooth 5.4, and is equipped with Alienware Command Center and AlienFX RGB backlit keyboard. It also has a 1TB M.2 PCIe NVMe SSD, 16GB DDR5 memory, and 12 hours of battery life. 2. Dell XPS 16 2024: This laptop comes pre-installed with Windows 11, has a 16-core Intel Core Ultra 7 processor, supports integrated Intel Iris Xe Graphics or NVIDIA GeForce 40 series graphics, has 32GB DDR5 memory and 512GB to 2TB of SSD storage. It also has 2 each of USB 3.2 Gen 1 and Gen 2, 2 Type-C (one of which supports Thunderbolt 4), 1 HDMI 2.1, 1 Ethernet, 1 headphone jack, and 1 microSD card slot. All of these laptops have high performance and are suitable for your needs. You can choose a laptop that suits you best based on your preferences and budget.

Figure 10: **A series of demos of Vision Search Assistant on novel images, events, and in-the-wild scenarios.** Vision Search Assistant delivers promising potential as a powerful multimodal engine.

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
