# OpenReview forum: "Vision Search Assistant: Empower Vision-Language Models as Multimodal Search Engines"
_ICLR.cc/2025/Conference — ICLR 2025 Conference Withdrawn Submission_

### Official Review · Reviewer_qL5v · 2024-10-29

**Soundness:** 2
**Presentation:** 2
**Contribution:** 2
**Rating:** 5
**Confidence:** 4

**Summary:**

This paper introduces a new framework called Vision Search Assistant, which aims to enhance the capabilities of mutimodal large language models and web agents through collaboration, targeting multimodal search tasks involving unfamiliar images and novel concepts. The authors identify three key aspects of visual search: what to search, how to search, and what to conclude. To address these, the proposed framework utilizes three steps: Visual Content Formulation, Chain of Search, and Collaborative Generation. The Vision Search Assistant is evaluated on the LLaVA-Wild dataset, and further assessed through open-set evaluations by human experts. Results indicate that Vision Search Assistant achieved the best outcomes across multiple metrics.

**Strengths:**

1. Open-domain visual search is valuable in practice, and combining multimodal large language models with search represents an intriguing research direction.
2. The paper identifies three critical questions in visual search — what to search, how to search, and what to conclude, which are interesting research points.

**Weaknesses:**

1. There is a lack of efficiency analysis, especially concerning the average latency per query, number of LMM calls, or token consumption. While efficiency is often an issue with large models, an essential analysis should not be overlooked.
2. The evaluation has some limitations. The paper only tests on the LLaVA-Wild dataset, which contains only 60 queries. In practice, many VQA datasets involving external knowledge, such as A-OKVQA, could serve as more comprehensive evaluation benchmarks.
3. There is insufficient information regarding reproducibility. Sampling parameters, such as temperature and top-p during large model inference, are not provided. Additionally, prompts used with large models (e.g., in Eq. 4 and Eq. 7) are not specified.

**Questions:**

1. What is the motivation for transforming visual regions into text descriptions for search, instead of directly using images? In my view, some critical information may be lost in the descriptive process, and many models struggle with entity recognition (such as the athlete’s name in Fig. 3), potentially affecting search accuracy. How do the authors think of this issue?
2. In Table 1, what does "LLava-1.6-7B (Baseline)" represent? Does it indicate zero-shot LLava? Why does the performance on Conversation decline when image search is introduced (naive search)? The way Google Image Search is utilized requires further explanation.
3. There is a lack of detailed qualitative analysis. The authors could provide a case study showing descriptions of each region, the results of each search chain step, and the final model-generated response to illustrate the benefits of the proposed method.
4. In Fig. 6, the authors conduct human evaluations of open-domain QA. What criteria are used to assess Factuality, Relevance, and Supportiveness? If these definitions follow prior work, citations should be provided; otherwise, detailed explanations are needed. Furthermore, are the human evaluation results statistically significant and consistent?

---

### Official Review · Reviewer_zrFM · 2024-10-31

**Soundness:** 2
**Presentation:** 3
**Contribution:** 2
**Rating:** 3
**Confidence:** 5

**Summary:**

The paper tackles the challenge of enabling vision-language models (VLMs) to understand and respond to unfamiliar visual content by integrating real-time web search capabilities.

The authors introduce the **Vision Search Assistant**, a framework that combines VLMs with web agents to perform open-world Retrieval-Augmented Generation, enhancing the model's ability to handle novel images and events.

Extensive experiments show that the Vision Search Assistant achieves good performance in both open-set and closed-set QA benchmarks.

**Strengths:**

The paper tackles the challenge of enabling vision-language models (VLMs) to understand and respond to unfamiliar visual content by integrating real-time web search capabilities. To this end, the authors introduce the **Vision Search Assistant**, a framework that combines VLMs with web agents to perform open-world Retrieval-Augmented Generation, enhancing the model's ability to handle novel images and events.

Here are Key Contributions:
  - **Visual Content Formulation**: Extracts and correlates object-level descriptions from images.
  - **Web Knowledge Search**: Uses an iterative algorithm to progressively gather and summarize relevant web knowledge.
  - **Collaborative Generation**: Integrates visual and textual information to generate accurate responses.

Extensive experiments show that the Vision Search Assistant improves the performance of the base model in both open-set and closed-set QA benchmarks.

**Weaknesses:**

### Method
1. **Complexity**: The approach is overly complicated, essentially trading time for performance.
2. **Correlated Formulations**: How are these extracted? The function \( F_{VLM} \) takes an image and two captions, so how can it output correlated formulations? Is \( F_{VLM} \) specifically trained for this task?

### Experiments
1. **Insufficient Experiments**: The proposed method is essentially retrieval-augmented generation (RAG), yet the authors barely discuss its relevance to existing RAG works. Furthermore, there are no experimental comparisons between the proposed method and state-of-the-art works, which is unacceptable in research. Given that New Bing (Copilot) is known as an excellent RAG system, it would be more convincing to compare with it.
2. **Impact of Open-Set Detectors**: How do open-set detectors affect the performance of the proposed method?
3. **Generalization Across VLLMs**: The proposed method appears to be training-free. Why were experiments only conducted on LLaVA 1.6? It is crucial to validate the generalization of this method across different VLLMs.
4. **Open-Set Benchmark**: What does the open-set benchmark look like? How is fairness ensured during subjective evaluation?
5. **Overclaim**: Why is the Chain of Search algorithm considered the answer to “How to search”? There are no quantitative results to support this claim. Additionally, the explanation for “By what to conclude” should be “How to conclude”. “What to conclude” focuses more on content selection.
6. **Efficacy of the Proposed Method**: The authors use many qualitative results to prove the efficacy of the method, which is not convincing. Sometimes, if VLLMs have never seen the objects before, the captions extracted from VLLMs make it difficult to retrieve results from the web.

### Additional Comments
- **Political Examples**: Using examples related to politics is not advisable. Many VLLMs avoid discussing such topics, which may lead to unfair implications about model performance.
- **Overall Assessment**: In my view, this paper is not ready for publication due to numerous weaknesses. I vote to reject this paper and hope the authors can further improve their work.

**Questions:**

Please see weaknesses.

---

### Official Review · Reviewer_YunS · 2024-11-03

**Soundness:** 2
**Presentation:** 2
**Contribution:** 2
**Rating:** 5
**Confidence:** 4

**Summary:**

To tackle the challenge of handling unseen images and novel concepts, this paper proposes a novel framework called Vision Search Assistant designed to empower vision-language models (VLMs) to function as multimodal search engines. This framework facilitates collaboration between VLMs and web agents, enabling the system to perform open-world RAG by leveraging the visual understanding capabilities of VLMs and the real-time information access of web agents.

**Strengths:**

1. This paper tackles the challenge of handling unseen images and novel concepts, which traditional VLMs struggle with due to their reliance on previously seen data.
2. This paper introduces an algorithm that allows for progressive summarization of web knowledge, enhancing the system's ability to handle novel and unseen data.
3. This paper tries to answer the remained questions and makes some ablation studies, such as what to search, how to search, and by what to conclude.

**Weaknesses:**

1. The introduction of the experimental section is somewhat careless. For instance, Figure 5 is provided in this paper, but there is no corresponding explanation or discussion of its content within the main text.
2. This paper lacks a detailed description of the open-set dataset used in the evaluation.
3. In Figure 6, the proposed method significantly outperforms GPT-4 web, but there is no clear explanation in the method section that justifies this improvement. Additionally, the paper does not analyze or discuss this result.
4. The experiments conducted are not sufficient, and this paper lacks comparisons with more open-world methods.
5. This paper does not provide information on the computational efficiency or time consumption of the proposed framework, particularly the Chain of Search algorithm, which is noted for its complexity.

**Questions:**

1. Is Vision Search Assistant allowed to access the internet during the Closed-set Evaluation? If so, does its search scope differ from that of other comparison models?
2. Why does the paper use textual descriptions of objects (line 424) instead of visual object representations for search?
3. How is the performance of the web knowledge search measured, and why is this aspect not explained in the paper?
4. How is the performance of the web knowledge search measured? This paper does not explain how the performance of the Web Knowledge Search component is measured.
5. If the web knowledge search is not accurate, how does this affect the final results returned by the system? What measures are in place to handle such inaccuracies?

---

### Official Review · Reviewer_m5rS · 2024-11-04

**Soundness:** 2
**Presentation:** 2
**Contribution:** 1
**Rating:** 3
**Confidence:** 5

**Summary:**

This paper proposes the Vision Search Assistant to address the issue that traditional methods struggle with understanding unfamiliar visual content. A workflow comprising three parts is proposed: Visual Content Formulation, Web Knowledge Search, and Collaborative Generation. Experiments on both open-set and closed-set QA tasks demonstrate that the model proposed in this paper outperforms other models significantly and can be widely applied to existing VLMs. The ablation studies validate each component of the model effectively.

**Strengths:**

1. This paper effectively addresses the challenges posed by traditional methods in comprehending unfamiliar visual content.
2. The experiment results achieved good performance.

**Weaknesses:**

1. The framework appears to be a typical application framework for LLMs that combines captioning, searching, and RAG. This seems to be a combination of A + B + C. The author needs to highlight their contributions.
2. Which search engine is being used? It doesn't seem to be mentioned in the paper.
3. The discussion of factuality, relevance, and supportiveness in the Open-Set Evaluation could be more detailed, as the current descriptions may not fully convey their specific meanings.
4. In the ablation study, providing quantitative results, such as in tabular form, would enhance the clarity and impact of the findings.
5. The experiments could benefit from a clearer explanation of the specific type of score being referenced, as this would help readers better understand the results.

**Questions:**

1. Could you provide more detailed explanations of the terms "actuality", "relevance", and "supportiveness" in the context of the Open-Set Evaluation? This might help clarify their specific meanings for the readers.
2. Would it be possible to include quantitative results in the ablation study, perhaps in tabular form, to enhance the clarity and impact of your findings?
3. Could you clarify the specific type of score being referenced in your experiments? This additional detail would help readers better understand the results.

---

### Note · Authors · 2024-11-13

I have read and agree with the venue's withdrawal policy on behalf of myself and my co-authors.